# Are there differences between COVID-19 and non-COVID-19 inpatient pressure injuries? Experiences in Internal Medicine Units

Leticia Nieto-García[1], Adela Carpio-Pérez[2‡], María Teresa Moreiro-Barroso[3‡], Emilia Ruiz-Antúnez[4‡], Ainhoa Nieto-García[1‡], Montserrat Alonso-Sardón[5]*

1 School of Nursing and Physiotherapy, University of Salamanca, Salamanca, Spain, 2 Internal Medicine Service, University Hospital of Salamanca, Institute for Biomedical Research of Salamanca (IBSAL), Tropical Disease Research Centre of the University of Salamanca (CIETUS), Salamanca, Spain, 3 Internal Medicine Service, University Hospital of Salamanca, Salamanca, Spain, 4 Training, Development and Innovation Area, University Hospital of Salamanca, Salamanca, Spain, 5 Preventive Medicine and Public Health Area, University of Salamanca, Institute for Biomedical Research of Salamanca (IBSAL), Tropical Disease Research Centre of the University of Salamanca (CIETUS), Salamanca, Spain

☉ These authors contributed equally to this work.
‡ ACP, MTMB, ERA and ANG also contributed equally to this work.
* sardonm@usal.es

**Data Availability Statement:** All relevant data are within the manuscript and its Supporting information files.

## Abstract

### Background

Pressure Injuries (PIs) are major worldwide public health threats within the different health-care settings.

### Objective

To describe and compare epidemiological and clinical features of PIs in COVID-19 patients and patients admitted for other causes in Internal Medicine Units during the first wave of COVID-19 pandemic.

### Design

A descriptive longitudinal retrospective study.

### Setting

This study was conducted in Internal Medicine Units in Salamanca University Hospital Complex, a tertiary hospital in the Salamanca province, Spain.

### Participants

All inpatients ≥18-year-old admitted from March 1, 2020 to June 1, 2020 for more than 24 hours in the Internal Medicine Units with one or more episodes of PIs.

**Funding:** No funding was received for any aspect of this study.

**Competing interests:** The authors have declared that no competing interests exist.

## Results

A total of 101 inpatients and 171 episodes were studied. The prevalence of PI episodes was 6% and the cumulative incidence was 2.9% during the first-wave of COVID-19. Risk of acute wounds was four times higher in the COVID-19 patient group (p<0.001). Most common locations were sacrum and heels. Among hospital acquired pressure injuries a significant association was observed between arterial hypertension and diabetes mellitus in patients with COVID-19 diagnosis.

## Conclusion

During the first wave of COVID-19, COVID-19 patients tend to present a higher number of acute wounds, mainly of hospital origin, compared to the profile of the non-COVID group. Diabetes mellitus and arterial hypertension were identified as main associated comorbidities in patients with COVID-19 diagnosis.

## 1. Introduction

Since the outbreak of the novel coronavirus (2019-nCoV) in December 2019 in Wuhan, China, confirmed cases have appeared in countries around the world with a serious impact in Spain, where the situational reports of World Health Organization (WHO) revealed 239 801 confirmed cases of COVID and 29 045 deaths as of June 2020 [1], period considered to be the first wave in this country.

National Pressure Injury Advisory Panel (NPIAP) points out that COVID-19 crisis has also brought significant changes in the implementation of preventive measures of pressure injuries (PI) [2], especially in the middle of the pandemic, since some patients spend prolonged stays in Intensive Care Unit (ICU) or other medical services. During the COVID-19 pandemic, several extrinsic factors associated with its propensity to overwhelm health care systems should be taken into consideration as risk factors of PI formation: lack for appropriate equipment (support surfaces, mattresses) or skin and wound care products, respiratory isolation and understaff of nursing professionals.

The last European Pressure Ulcer Advisory Panel (EPUAP) Virtual Meeting in September 2020 [3] highlighted etiology factors linked to development of PIs such as COVID-19 virus pathophysiology (systemic coagulopathy, hypercoagulation, microvascular occlusion) [4–7], the role of inflammation, limited reposition caused by hemodynamic instability or profound hypoxia, use of prone position as adjuvant therapy and the increase in the use of medical device-related PIs (tracheostomy tubes, feeding tubes and oxygen delivery devices). Finally, survivors of severe cases have required prolonged recovery hospital stay due to severe weakness, comorbidities related to COVID-19 or chronic diseases.

We performed a single institutional study in hospitalized patients who developed one or more episodes of PIs during the first wave of the COVID-19 pandemic in the Internal Medicine Units to describe and compare epidemiological and clinical features of acute or chronic PIs in COVID-19 patients and patients admitted for other causes.

## 2. Material and methods

### 2.1. Study design, setting and data source

A descriptive longitudinal retrospective study was performed at the Salamanca University Hospital Complex *(Complejo Asistencial Universitario de Salamanca "CAUSA", in Spanish)*, Spain. This is a public and tertiary hospital with 903 acute beds, 110 medium-long stay beds and 45 hospital medical services. It provides healthcare cover to 331.048 inhabitants (Salamanca population January 1, 2020; INE: https://www.ine.es/).

We defined the first wave of COVID-19 pandemic in Spain as the period from March 1, 2020 to June 1, 2020. A total of 4289 individuals were confirmed COVID-19 cases in the Salamanca Health Area. Of these, 1196 cases were hospitalized in the Internal Medicine Units of Salamanca University Hospital and 363 died [8]. The need for beds due to the high flow of patients admitted during the study period turned many Units into COVID-19 wards.

This study followed the Strengthening the Reporting of Observational Studies in Epidemiology (STROBE) reporting guideline. Two investigators/researchers reviewed the Nursing Care Reports (NCR) and Clinical Discharge Reports (CDR) records of admitted subjects who had one or more PIs during their hospital stay at the Internal Medicine Units of the Salamanca University Hospital Complex, from March 1, 2020 to June 1, 2020.

The care management program GACELA-Care® was used for data collection from NCR, extracting variables related to the characteristics of PIs, their carried-out treatment and the products used. GACELA-Care is a software that allows the registration of nursing actions in the care and monitoring of the hospitalized patient. This Health Information System uses the *"episode of care"* as the unit of record, defined as *"the process of care for an illness or demand made by the patient, which begins with the first contact with the health services and ends with the last contact related to the specific episode"*. Next, the CDR corresponding to the previously selected episodes were reviewed to complete the patient´s clinical, demographic and administrative information. In addition, the CDR allowed to fill in data not registered in Gacela Care®, especially for those patients who came from a previous stay in another hospital unit.

### 2.2. Eligibility criteria

According to National Pressure Ulcer Advisory Panel (NPUAP), we considered as PI any localized damage to the skin and underlying soft tissue, usually over a bony prominence or related to a medical or other device, result of intense and/or prolonged pressure or pressure in combination with shear [9].

**2.2.1. Inclusion criteria.** All adult patients (≥18 year old) admitted from March 1, 2020 to June 1, 2020 for more than 24 hours to Internal Medicine Units and diagnosed of PI in their medical or nursing records.

**2.2.2. Exclusion criteria.** Pediatric population and skin wounds registered as vascular, surgical, traumatic, neoplastic, Moisture-Associated skin damage (MASD) and other type of skin wounds, plus non-COVID-19 cases that were admitted to the hospital prior to March 1, 2020.

### 2.3. Data collection

The variables were classified into 2 groups: a) variables that allow characterizing the sample of patients, such as gender, age, hospitalization unit, principal diagnosis and secondary diagnosis/other diagnosis, reason for discharge, days of hospital stay, exitus letalis. b) variables related to the specific characteristics of PIs, such as number of PIs per patient, location, ulcer origin (domicile, unit or ICU acquired PI or Primary Health Care), date of appearance/registration

and closing date of the process, pain related to PI, stage according to the NPUAP staging system [stage I, Nonblanchable erythema of intact skin; stage II, partial-thickness skin loss involving the epidermis or dermis; stage III, full-thickness skin loss that may extend to, but not through, the fascia; stage IV, full-thickness skin loss involving deeper structures, such as muscle, bone or joint structures] [9], PI characteristics (size, shape, exudate type, edges, perilesional skin), infection and microorganism found in culture. Infection diagnoses are based on the indicators of suspected local infection enclosed on NPUAP/NPIAP/PPPIA quick guide [10], such as the increased exudates or change in its nature, pain or surrounding tissue temperature, larger size or depth, presence of pocketing/bridging, foul odor, friable granulation tissue, delayed healing and necrotic tissue. Samples for culture were taken by the nursing team or physicians from PI showing signs of infection by superficial swab or aspiration.

Data were collected and analyzed by two analysts. Patients were grouped according to the principal diagnosis: patients with a diagnosis of laboratory-confirmed SARS-CoV-2 infection (hereinafter referred to as "COVID-19 patients") and patients admitted for other causes (hereinafter referred to as "non-COVID-19 patients"). Skin injury episodes were stratified and analyzed into acute *versus* chronic pressure injuries.

### 2.4. Statistical analyses

We displayed the patients or PI characteristics using descriptive statistics: numbers (n), percentages (%), mean or median, and standard deviation (SD) or interquartile range (IQR = $Q_3$-$Q_1$). The Shapiro-Wilk test was used to verify normality. In the bivariate analysis, a Chi-square ($\chi^2$) test was used to compare the association between demographics and clinical categorical variables of patients with and without COVID groups and the measured outcome was expressed as the odds ratio (OR) together with the 95% CI for OR. Continuous variables were compared with Student's t-test or Mann-Whitney test for two groups, depending on their normal or non-normal distribution. ANOVA allowed us to analyze the influence of independent nominal variables on a continuous dependent variable. In the second stage of the analysis, a post-hoc test (Fisher's Least Significant Difference (LSD), Bonferroni Test) was used after we found a statistically significant result and needed to determine where our differences truly came from. A p-value of $p < 0.05$ was considered statistically significant. All statistical tests were performed using SPSS software (Statistical Package for the Social Sciences) version 26.0.

### 2.5. Ethics statement

The database supporting the findings of this study is available from the corresponding author on reasonable request. The study was approved by the Ethics Committee of the University Hospital of Salamanca (Code: PI 2019 03 208) (see S1 File). All data were kept confidential and processed anonymously in accordance with the requirements of Law 3/2018 of 5 December on the Protection of Personal Data and guarantee of digital rights.

## 3. Results

### 3.1. Frequency of PIs in the Internal Medicine Units

From March 1 to June 1, 2020, a total of 4286 episodes were recorded at the Salamanca University Hospital. Of them, 2857 episodes were recorded in Internal Medicine Units. Finally, 171 PI episodes requiring nursing care were recorded in the Internal Medicine Units during the first-wave COVID-19 pandemic (see flow diagram, Fig 1); 228 PI episodes were recorded during the same time period in the year prior to the pandemic (from March 1, 2019 to June 1, 2019). These 171 PI episodes corresponded to 101 patients, 86 episodes corresponded to 51

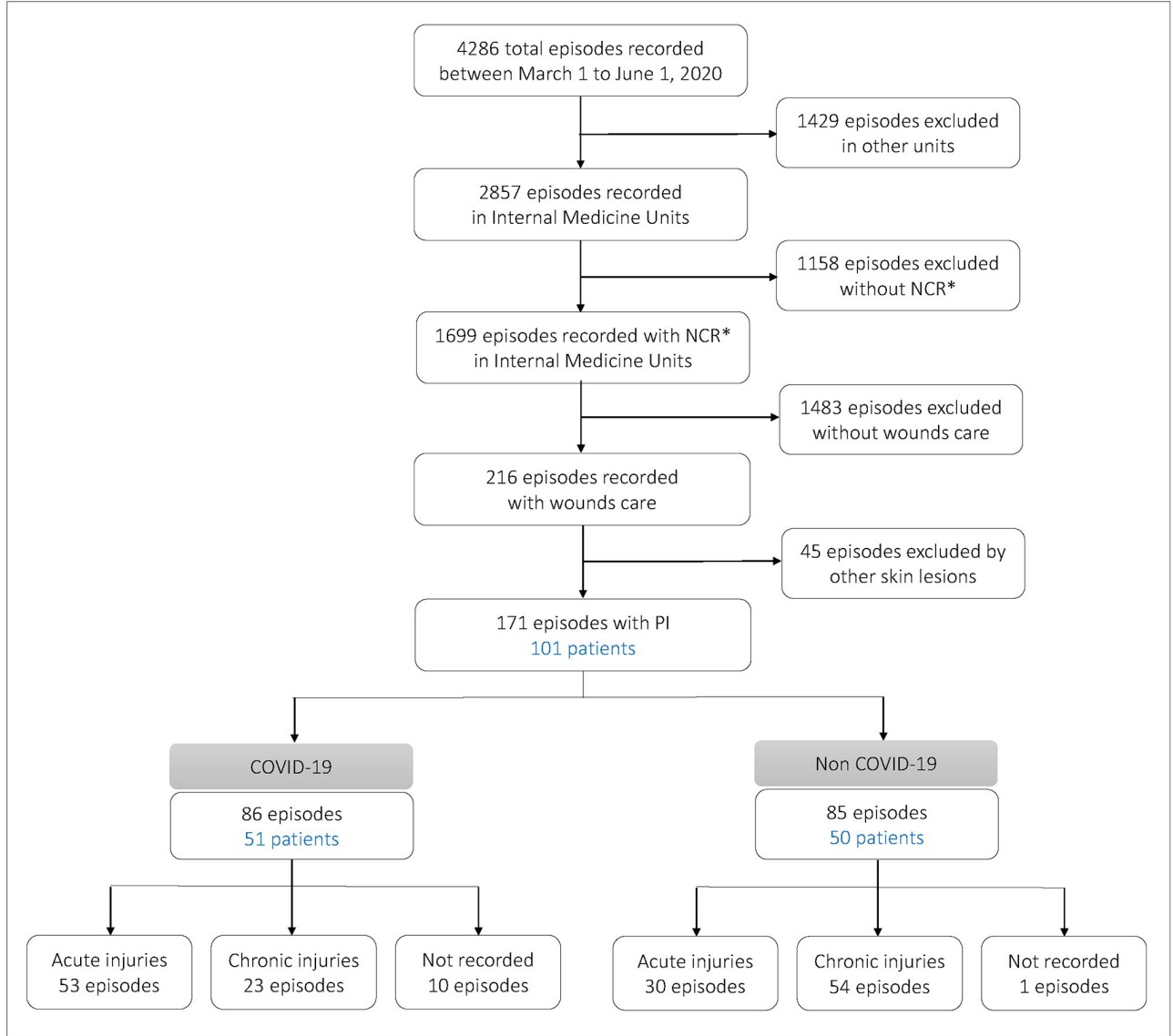

**Fig 1. Flow diagram to show episodes recorded between March 1 and June 1, 2020.** *NCR: Nursing Care Reports.

COVID-19 patients and 85 episodes to 50 "non-COVID-19 patients". Only 2 patients required hospital readmission during the study period for SARS-CoV-2 transmission during their first hospital stay. The proportion of COVID-19 patients with PI in the Internal Medicine Units of Salamanca University Hospital during the first-wave COVID-19 pandemic was 4.3% (51/1196).

The *prevalence* of PI episodes in Internal Medicine Units of Salamanca University Hospital during the first-wave COVID-19 pandemic was 6% (171/2857). Of these, 77 episodes corresponded to injuries before their hospital admission, while 83 episodes were injuries that occurred during their hospital stay (11 episodes the site and time of appearance is unknown). Thus, the *cumulative incidence* of PI episodes in the Internal Medicine Units of Salamanca University Hospital during the first-wave COVID-19 pandemic was 2.9% (83/2857).

**Table 1. Patient cohort characterization.**

| Variables | Total N = 101 | COVID-19 (N₁ = 51) | Non-COVID-19 (N₂ = 50) | |
|---|---|---|---|---|
| **Gender** | n (%) | n (%) | n (%) | p-value* |
| Male | 45 (44.6) | 24 (47.1) | 21 (42.0) | 0.609 |
| Female | 56 (55.4) | 27 (52.9) | 29 (58.0) | |
| **Age, years** | | | | |
| <60 years | 1 (1.0) | 1 (2.0) | - | 0.248 |
| 60–69 years | 14 (13.9) | 11 (21.6) | 3 (6.0) | |
| 70–79 years | 17 (16.8) | 8 (15.7) | 9 (18.0) | |
| 80–89 years | 40 (39.6) | 19 (37.3) | 21 (42.0) | |
| 90–99 years | 27 (26.7) | 11 (21.6) | 16 (32.0) | |
| 100 years | 2 (2.0) | 1 (2.0) | 1 (2.0) | |
| Mean ± SD | 82.9±10.8 | 81.1±12.3 | 84.8±8.7 | 0.080 |
| **No. episodes per patient** | | | | |
| 1 | 60 (59.4) | 28 (54.9) | 32 (64.0) | 0.162 |
| 2 | 22 (21.8) | 15 (29.4) | 7 (14.0) | |
| 3 or more | 19 (18.8) | 8 (15.7) | 11 (22.0) | |
| Mean ± SD | 1.8±1.3 | 1.8±1.2 | 1.8±1.4 | 0.988 |
| **Principal diagnosis** | | | | |
| COVID-19 or SARS-CoV-2 infection | 51 (50.5) | 51 (100.0) | - | |
| Sepsis/septic shock/bacteremia | 17 (16.8) | - | 17 (34.0) | |
| Pneumonia/respiratory tract infection, . . . | 12 (11.9) | - | 12 (24.0) | |
| Urinary tract infection | 7 (6.9) | - | 7 (14.0) | |
| Neoplasm diagnosis | 5 (5.0) | - | 5 (10.0) | |
| Chronic renal disease | 5 (5.0) | - | 5 (10.0) | |
| Heart failure | 3 (3.0) | - | 3 (6.0) | |
| Digestive disease | 1 (1.0) | - | 1 (2.0) | |
| **Stay in Intensive Care Unit** | 12 (11.9) | 8 (15.7) | 4 (8.0) | 0.233 |
| **Case Fatality Rate** | 15 (14.9) | 7 (13.7) | 8 (16.0) | 0.748 |
| **Hospital stay (days),** mean ± SD | 21.5±14.0 | 27.3±13.7 | 15.7±11.8 | <0.001* |

*p-values with statistical significance level of 5% (p <0.05).

## 3.2. Patient characteristics

Of the total 101 patients with PI episodes recorded in the Internal Medicine Units during the first-wave COVID-19 pandemic, 45 (44.6%) were men and 56 (55,4%) were women. The mean (±SD) age for the overall cohort was 82.9 years (±10.8) [median (IQR), 84 years (90–77)], range (48 a 100). Principal diagnosis was COVID-19 or SARS-CoV-2 infection in 51 patients; other principal diagnoses were sepsis/septic shock/bacteremia (17 patients) and pneumonia/respiratory tract infection (12 patients).

Table 1 summarizes the main characteristics of the sample. Percentage of females (58% vs. 52.9%; p = 0.609) and mean age were slightly higher in non-COVID-19 patients (84.8±8.7 vs. 81.1±12.3; p = 0.080), with no statistically significant differences between groups. The average hospital stay was significantly longer in the COVID-19 patient group (27.3±13.7 vs. 15.7±11.8; p<0.001). Twelve (11.9%) patients required a stay in an ICU, higher risk in the COVID-19 patient group (OR = 2.1; 95% CI, 0.6–7.6; p = 0.233); and 15 (14.9%) patients died, there were no significant differences between the two groups (0.748).

### 3.3. PI episodes characteristics

Of the total of 171 PI episodes recorded in the Internal Medicine Units during the first-wave COVID-19 pandemic, 83 episodes were injuries that occurred during the hospital stay (acute injuries), also known as Hospital Acquired Pressure Injuries (HAPI), and 77 episodes corresponded to injuries before the hospital admission (chronic injuries), 11 episodes the site and time of appearance is unknown. Risk of acute wounds was four times higher in COVID-19 patient group (OR = 4.1; 95% CI, 2.1–8.0; p<0.001) (see Fig 2).

Fig 3 shows the anatomic location of PIs in COVID-19 and non-COVID-19 patients. The most common locations were sacrum and heels with similar percentages.

Table 2 describes main PI characteristics (shape, exudate, edges, tunneling. . .) in two-episode groups.

**3.3.1. Acute injuries.** Of the 83 episodes, 53 occurred in patients with a COVID diagnosis and 30 in patients admitted for other diagnoses. In both subgroups, most PIs were classified between stage I and II (90.6% in COVID-19 vs 86.7% non-COVID-19). Only one patient with HAPI in the COVID-19 group required evaluation and follow-up from Plastic Surgery or Dermatology, compared to three patients in non-COVID-19 group. A percentage of the group of acute wounds (16.8%) used exclusively products aimed at prevention, such as Hyperoxygenated Fatty Acid (HOFA), protective dressing or heel pad. Only 21.6% required debridement: enzymatic (n = 14) and surgical (n = 4). Regarding other types of treatments, 27.8% needed moist environment dressing.

In the first stage of the analysis, the comorbidity showed statistically significant differences between acute injury episodes in patients with COVID-19 and non-COVID-19 (p<0.001). In the second stage of the analysis, we found statistically significant results between pairs of groups as Fig 4 illustrates. We noted a significant association in HAPIs between the presence of arterial hypertension (OR = 6.9; 95% CI, 1.9–25.6; p = 0.002) and diabetes mellitus

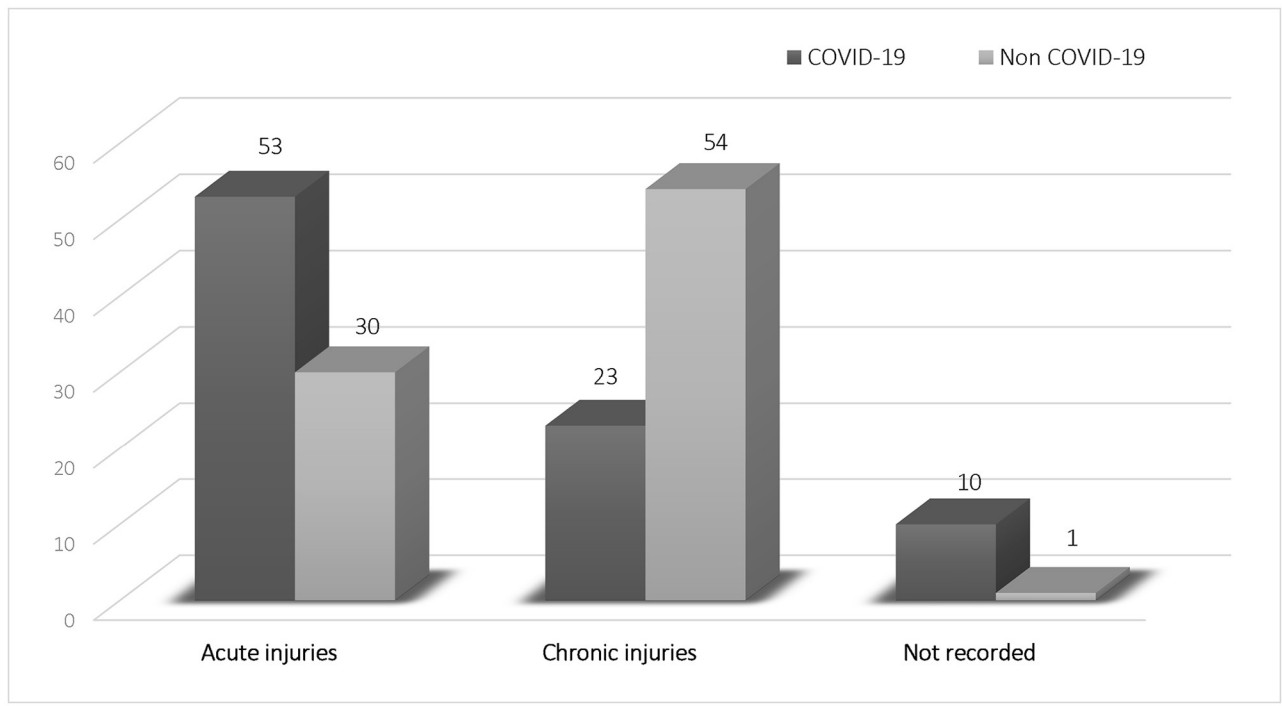

**Fig 2. Acute vs. chronic PI episodes in COVID-19 and non-COVID-19 patients.**

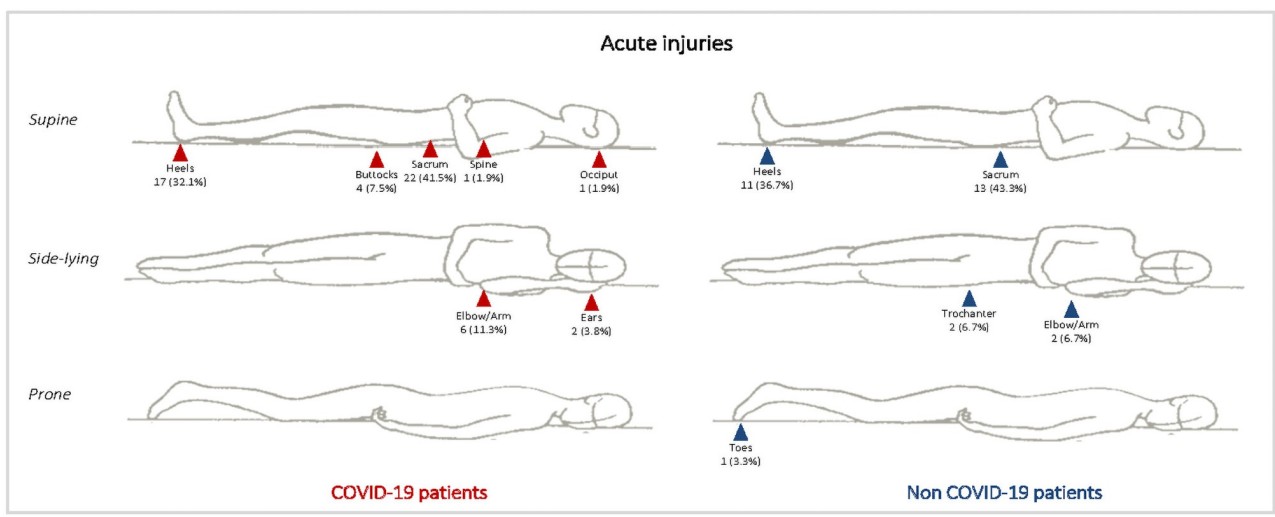

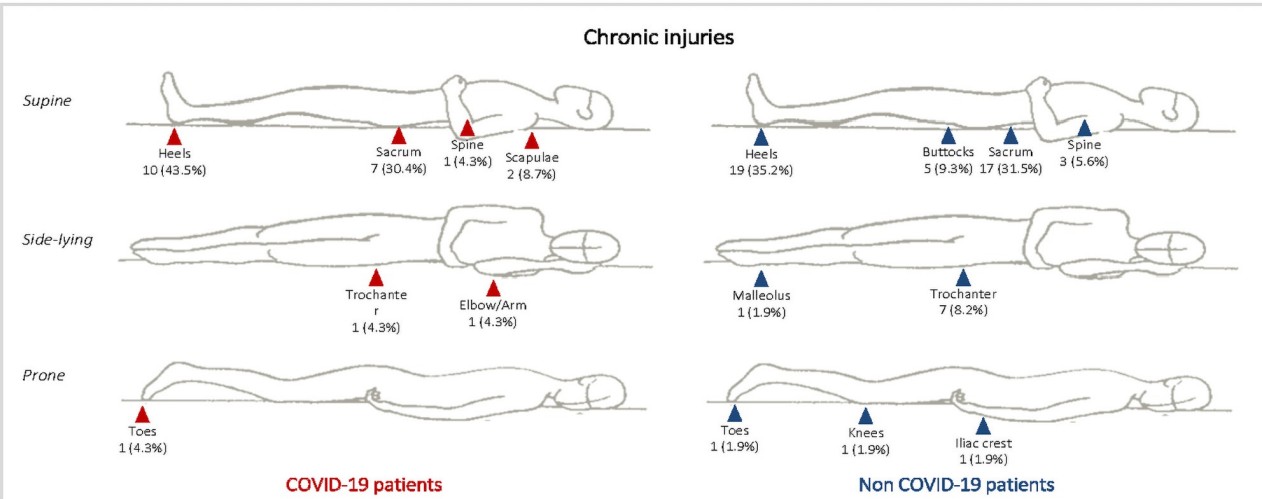

**Fig 3. Anatomic location of acute vs. chronic PIs in COVID-19 and non-COVID-19 patients.**

(OR = 5.5; 95% CI, 1.2–26.1; p = 0.019) in patients with COVID-19 diagnosis. On the other hand, non-COVID-19 patients were more associated to other comorbidities as oncologic processes or neurodegenerative/cognitive impairment.

12% of the episodes of HAPIs (6 in COVID-19 group vs 4 non-COVID-19) were diagnosed with infection after displaying clinical signs, and culture was obtained in four of them, all belonging to COVID-19 group, with the following outcomes: *Candida Albicans* (1), *Pseudomonas Aeruginosa* (1), *Streptococcus Faecium* (1), *Pseudomonas Aeruginosa and Streptococcus Faecium* (1).

In COVID-19 patients, HAPIs infection was associated in a higher percentage with patients with malnutrition (50%) and kidney diseases (66.7%); in the group of patients without COVID-19 diagnosis, infected HAPI was associated in a higher proportion with the presence of cardiovascular, chronic pulmonary diseases and kidney comorbidity (75% in all cases): although these results were not statistically significant.

**3.3.2. Chronic injuries.** Of the 77 episodes, 23 occurred in patients with a COVID diagnosis and 54 in patients admitted for other diagnoses (only 1 episode the site and time of

**Table 2. Acute vs. chronic injuries features in COVID-19 and non-COVID-19 patients.**

| Variables | Acute injuries (N₁ = 83) | | | Chronic injuries (N₁ = 77) | | |
|---|---|---|---|---|---|---|
| | COVID-19 (N = 53) | Non-COVID-19 (N = 30) | | COVID-19 (N = 23) | Non-COVID-19 (N = 54) | |
| **Stage** | n (%) | n (%) | p-value* | n (%) | n (%) | p-value* |
| I | 18 (34.0) | 12 (40.0) | 0.198 | 4 (17.4) | 10 (18.5) | 0.033* |
| II | 30 (56.6) | 14 (46.7) | | 12 (52.2) | 20 (37.0) | |
| III | 2 (3.8) | 4 (13.3) | | 7 (30.4) | 9 (16.7) | |
| IV | 3 (5.7) | - | | - | 15 (27.8) | |
| **Shape** | | | | | | |
| Oval | 10 (18.9) | 10 (33.3) | 0.624 | 8 (34.8) | 14 (25.9) | 0.079 |
| Round/circular | 8 (15.1) | 4 (13.3) | | 1 (4.3) | 7 (13.0) | |
| Irregular | 4 (7.5) | 4 (13.3) | | 1 (4.3) | 14 (25.9) | |
| *Not recorded* | *31 (58.5)* | *12 (40.0)* | | *13 (56.5)* | *19 (35.2)* | |
| **Wound edges** | | | | | | |
| Delimited | 9 (17.0) | 8 (26.7) | 0.542 | 6 (26.1) | 11 (20.4) | 0.480 |
| Diffuse/Indistinguishable | 1 (1.9) | 1 (3.3) | | 1 (4.3) | 6 (11.1) | |
| Damaged | 4 (7.5) | 1 (3.3) | | 3 (13.0) | 4 (7.4) | |
| *Not recorded* | *39 (73.6)* | *20 (66.7)* | | *13 (56.5)* | *33 (61.1)* | |
| **Wound exudate** | | | | | | |
| None/dry wound | 8 (15.1) | 9 (30.0) | 0.522 | 1 (4.3) | 11 (20.4) | 0.493 |
| Sanguineous | 2 (3.8) | - | | - | 2 (3.7) | |
| Serous | 4 (7.5) | 2 (6.7) | | 1 (4.3) | 2 (3.7) | |
| Purulent | 1 (1.9) | - | | 2 (8.7) | 5 (9.3) | |
| Undetermined exudate | 1 (1.9) | 1 (3.3) | | - | 4 (7.4) | |
| *Not recorded* | *37 (69.8)* | *18 (60.0)* | | *19 (82.6)* | *30 (55.6)* | |
| **Perilesional skin** | | | | | | |
| Erythema | 5 (9.4) | 4 (13.3) | 0.563 | 3 (13.0) | 9 (16.7) | 0.327 |
| Maceration | - | - | | 1 (4.3) | 4 (7.4) | |
| Desquamation | 1 (1.9) | 1 (3.3) | | 1 (4.3) | - | |
| Excoriation | - | - | | - | 2 (3.7) | |
| Edema | 1 (1.9) | - | | - | 2 (3.7) | |
| Lacerated skin | - | 1 (3.3) | | - | - | |
| *Not recorded* | *46 (86.8)* | *24 (80.0)* | | *18 (78.3)* | *37 (68.5)* | |
| **Undermining/Cavitated PI** | - | 1 (3.3) | 0.181 | 1 (4.3) | 3 (5.5) | 0.827 |
| **Pain** | 6 (11.3) | 6 (20.0) | 0.280 | 3 (13.0) | 10 (18.5) | 0.557 |
| **Infection** | 6 (11.3) | 4 (13.3) | 0.787 | 3 (13.0) | 13 (24.1) | 0.275 |
| **Management strategies** | | | | | | |
| Only preventive products | 7 (13.2) | 7 (23.3) | 0.237 | 1 (4.3) | 1 (1.9) | 0.529 |
| Moist environment treatment | 16 (30.2) | 7 (23.3) | 0.503 | 8 (34.8) | 24 (44.4) | 0.431 |
| Debridement | 12 (22.6) | 6 (20) | 0.900 | 12 (52.2) | 19 (35.2) | 0.040* |
| Enzymatic Debridement | 10 | 4 | | 10 | 13 | |
| Surgical debridement | 2 | 2 | | 2 | 6 | |
| Plastic surgery | 1 (1.9) | 3 (10.0) | 0.097 | 5 (21.7) | 3 (5.6) | 0.033* |

*p-values with statistical significance level of 5% (p <0.05).

appearance is unknown). There are statistically significant differences in staging; with a distribution of 45.5% of PIs in stage III-IV in non-COVID-19 group versus 30.4% in stage III of the COVID-19 group (without any PI recorded in stage IV). As shown Fig 4, same as the group with acute injuries, there is a significant association in the chronic PIs between diabetes

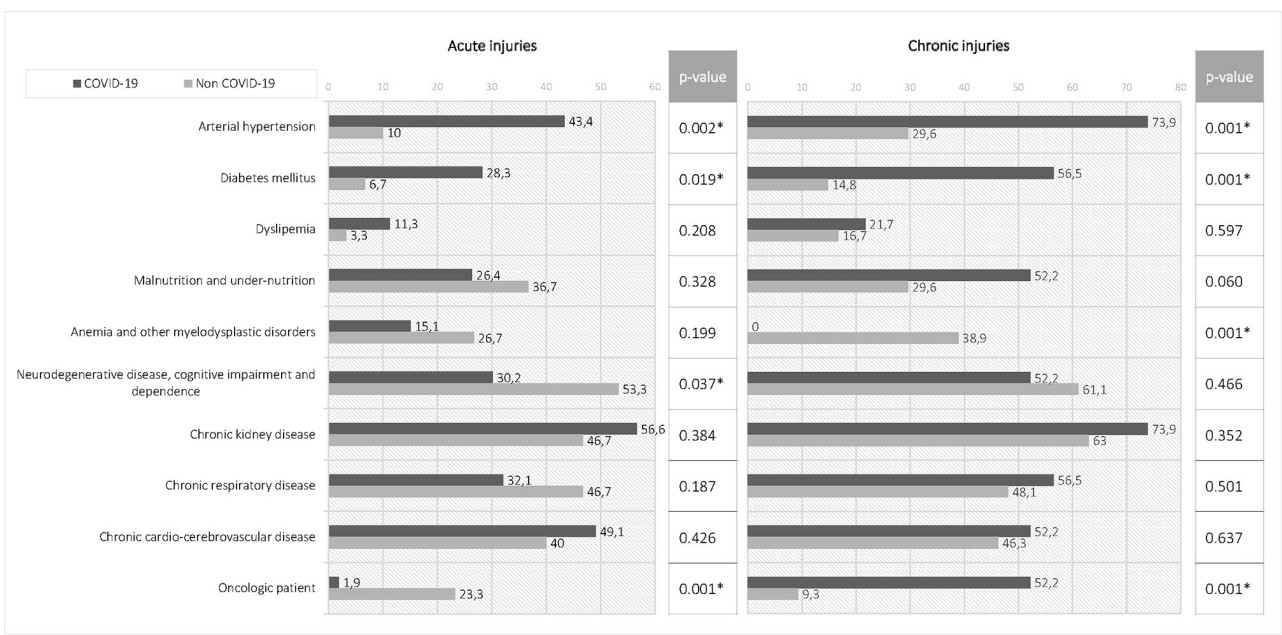

**Fig 4. Percentage distribution comorbidity in acute injuries vs. chronic injuries and p-values obtained for the observed difference between the paired samples.**

mellitus (OR = 7.4; 95% CI, 2.4–23.8; p = 0.001) and arterial hypertension (OR = 6.7; 95% CI, 2.2–20.2; p = 0.001) in COVID-19 patients. Five patients in the COVID-19 group and three in non-COVID-19 group required the attention of Plastic Surgery or Dermatology service. Debridement was performed in 40.2% of PIs; surgical debridement was more frequent in non-COVID-19 group. 41.5% of registered HAPIs required a moist environment dressing.

Among chronic PIs, 20.7% of the skin lesions were infected (3 in COVID-19 group vs 13 non-COVID-19). The diagnosis of infection in the PIs was confirmed by microbiological analysis in nine episodes. The culture results in the COVID-19 group were *Escherichia Coli* (1), *Escherichia Coli + Morganella Morganii* (1) and *Pseudomonas Aeruginosa + Streptococcus Haemolyticus + Enterococcus Faecium + Candida Tropicalis* (1); and in the non-COVID-19 cultures were *Escherichia Coli* (1), *Morganella Morganii* (1), *Pseudomonas Aeruginosa* (1), *Proteus Mirabilis + Escherichia Coli* (1), *Proteus Mirabilis + Escherichia Coli + Pseudomonas Aeruginosa* (1) and *Serratia* (1).

In COVID-19 patients, infection of chronic PI was associated with cardiovascular pathology, kidney disease, arterial hypertension, anemia or other myelodysplastic disorders (100% in all cases) and diabetes mellitus (66.7%); in contrast, in the group of patients without COVID-19 diagnosis, infections were associated with the presence of kidney disease (69.2%), dementia and other neurological disorders with high degree of dependence (61.5%) and chronic pulmonary disease (61.5%); although these results were not statistically significant (p>0.05).

It should be noted that seven patients with chronic wounds and negative COVID-19 diagnosis were admitted due to complications associated with PI infection: three patients were admitted with the principal diagnosis of "cutaneous sepsis" or "soft tissue infection related to PI", one patient required readmission due to complications from a previous principal diagnosis of "urinary and respiratory sepsis and soft tissue infections" and was admitted with a secondary diagnosis of colonized sacral PI, one was admitted with main diagnosis "sacral PI

infection by *Morganella Morganii"* and finally, another patient was diagnosed with pararectal abscess in sacral PI.

## 4. Discussion

As the literature shows Hospital Acquired Pressure Injury (HAPI) is a well-known indicator of the quality of care in acute settings. A recent systematic review estimates that the average prevalence of PI among published studies in Europe is 10.8% with values ranging from 4.6% to 27.2% [11]. The 5[th] prevalence study carried out in Spain [12] estimated a prevalence of 7% with higher prevalence in services such as palliative care, intensive care and post-surgical and reanimation units. The study also indicates that 72.2% of PIs are of nosocomial origin, occurring in hospitals or nursing homes. The prevalence of PIs in this study (6%) is lower compared to previous studies, although the clinical context must be taken into account; the mean age of the sample was high and presented multiple combined pathologies. In both groups, the main comorbidities were associated with chronic illness, typical of the high mean age of our sample, such as diabetes mellitus, arterial hypertension, neurodegenerative diseases, kidney diseases, cerebrovascular diseases or respiratory diseases. All this leads to a clinical situation that may predispose to the development of PIs [13–15]. However, the two factors with the greatest association with the development of a HAPI in COVID-19 patients were diabetes mellitus and arterial hypertension. This association may be due to the established relationship between increased rates of hospital admission, severity and mortality of COVID-19 in patients with these comorbidities [16–20].

Statistical differences were observed with longer hospital stays in COVID-19 group. This may be explained by two possible reasons: first, the relationship of moderate or severe course of COVID-19 with longer hospital stays; second, by the effort to provide hospital beds for expected COVID-19, discharging non-COVID-19 patients. Our outcomes show statistically significant differences in terms of appearance time. In the COVID-19 group, HAPIs were more frequent than in the non-COVID-19 group, where more than half of patients had developed a PI prior to hospitalization.

Our study found no significant differences in terms of location of PIs acquired during the hospital stay, the heel area and sacral region were the most frequent with very similar percentages between the COVID-19 and non-COVID-19 group. However, the COVID-19 group recorded two acute PIs in the ear and one in the occiput compared to none in the non-COVID-19 group. The presence of HAPIs at these anatomic locations may be due to the fact that the respiratory status of hospitalized COVID-19 patients, especially critically ill, often interferes with standard preventive measures of repositioning patients. Prone positioning may be effective in improving respiratory status although some articles associate this position with an increased risk of facial HAPIs [21–24].

Despite COVID-19 patients presenting factors such as respiratory isolation, prohibition of family accompanent, lack of personal protective equipment for the health workforces for safe access to patient care, etc., which may have altered the implementation of the protocolized preventive measures, fewer PIs were observed compared to the same period of the previous year. This may sound contradictory after taking into account the serious clinical status of the patients, the high mean age of the sample and the presence of multiple chronic conditions. Authors consider that given the situation of care overload, the high number of hospital admission and quick restructuring of nursing staff in a short period caused an underreporting of PIs, both hospital origin and pre-existing before admission, due to the prioritization of health care over the registry. This is also reflected in the lack of registration of the specific characteristics of each PI such as depth, size, exudate, etc.–only variables such as staging and location were

completed in all episodes–although their assessment, management and decision making of their treatment was likely carried out in situ during the assistance practice.

## 4.1. Strength and limitations

Retrospective data collection together with the worst period of hospital overload may have led to an underestimation of the prevalence or incidence rates because a proportion of PIs were not documented or lacked care information. Regarding the diagnosis of infection, the latest recommendations consider deep tissue biopsy cultures as the reference standard for the microbiological analysis of PIs, however, in routine clinical practice of our department, the aspiration or superficial swab culture are still used more frequently because they are simpler, cheaper and less invasive methods.

Despite these limitations, this study provides the first analysis of the situation of PIs care during the worst moment of the pandemic in Spain. Therefore, the main strength is focused on its originality and novelty due to the lack of knowledge about the burden of the novel COVID-19 disease in the care process. This study may provide the basis for future research and comparisons related to the impact of COVID-19 on hospital care, as well as, helping to improve the protocols for the prevention, registration and treatment of PIs in future pandemic situations.

## 5. Conclusions

The study shows that HAPIs were more frequent in COVID-19 group during the first wave of COVID-19. Diabetes mellitus and arterial hypertension were identified as main associated comorbidities in patients with COVID-19 diagnosis. Evidence reported in this study once again supports the importance of appropriate preventive measures to avoid this complication.

## Supporting information

**S1 File. Ethics committee approval.**
(PDF)

**S1 Data.**
(SAV)

**S2 Data.**
(SAV)

## Author Contributions

**Conceptualization:** Leticia Nieto-García, Montserrat Alonso-Sardón.

**Data curation:** Leticia Nieto-García, Adela Carpio-Pérez, María Teresa Moreiro-Barroso, Emilia Ruiz-Antúnez, Ainhoa Nieto-García.

**Formal analysis:** Montserrat Alonso-Sardón.

**Funding acquisition:** Ainhoa Nieto-García.

**Investigation:** Leticia Nieto-García, Adela Carpio-Pérez, María Teresa Moreiro-Barroso, Emilia Ruiz-Antúnez, Montserrat Alonso-Sardón.

**Methodology:** Montserrat Alonso-Sardón.

**Software:** Leticia Nieto-García, Montserrat Alonso-Sardón.

**Supervision:** Montserrat Alonso-Sardón.

**Writing – original draft:** Leticia Nieto-García, Montserrat Alonso-Sardón.

**Writing – review & editing:** Leticia Nieto-García, Adela Carpio-Pérez, Montserrat Alonso-Sardón.

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
