## [Decision Letter · Decision Letter 0]

24 Sep 2021

PONE-D-21-24374Are there differences between COVID-19 and non-COVID-19 inpatient pressure injuries? Experiences in Internal Medicine UnitsPLOS ONE

Dear Dr. Alonso-Sardón,

Thank you for submitting your manuscript to PLOS ONE. After careful consideration, we feel that it has merit but does not fully meet PLOS ONE’s publication criteria as it currently stands. Therefore, we invite you to submit a revised version of the manuscript that addresses the points raised during the review process.

 Both reviewers' comments are attached below. Both had differing recommendations whether to accept the paper or not. However, both are in agreement that there is significant bias due to patient selection. And both have suggestions on how to improve this manuscript. Since methodology is the major set back of this submission, it could be that the right choice should be to remove this submission, rewrite the paper focusing on the correct patient population. The reviewers seem to prefer the acute pressure injuries as the group to focus on. I still have my bias in also including the chronic wounds in a separate analysis, focusing on issues that were not even mentioned, such as how many of those with deep wounds were offered an operation to alleviate their problem. I leave it up to the authors to make a decision whether to retreive their paper and submit a revised paper focusing on a more limited homogenous group, or to try and revise their paper taking advantage of the suggestions made by both reviewers.  Please submit your revised manuscript by Nov 08 2021 11:59PM. If you will need more time than this to complete your revisions, please reply to this message or contact the journal office at plosone@plos.org. Please include the following items when submitting your revised manuscript:A rebuttal letter that responds to each point raised by the academic editor and reviewer(s). You should upload this letter as a separate file labeled 'Response to Reviewers'.A marked-up copy of your manuscript that highlights changes made to the original version. You should upload this as a separate file labeled 'Revised Manuscript with Track Changes'.An unmarked version of your revised paper without tracked changes. You should upload this as a separate file labeled 'Manuscript'.

We look forward to receiving your revised manuscript.

Kind regards,

Itamar Ashkenazi

Academic Editor

PLOS ONE

Reviewers' comments:

Reviewer's Responses to Questions

**Comments to the Author**

1. Is the manuscript technically sound, and do the data support the conclusions?

Reviewer #1: Yes

Reviewer #2: No

2. Has the statistical analysis been performed appropriately and rigorously? 

Reviewer #1: I Don't Know

Reviewer #2: Yes

3. Have the authors made all data underlying the findings in their manuscript fully available?

Reviewer #1: Yes

Reviewer #2: Yes

4. Is the manuscript presented in an intelligible fashion and written in standard English?

Reviewer #1: Yes

Reviewer #2: Yes

5. Review Comments to the Author

Reviewer #1: Pressure injuries (PI) are a global health issue. The authors describe and compare epidemiological and clinical features of PIs in

COVID-19 patients and patients admitted due to other causes in Internal Medicine Units during the first wave of COVID-19 pandemic.

This study has a substantial inherent bias, performed during the COVID-19 pandemic, that may interfere with the main results. Though it was mentioned in the study limitations, it should be elaborated more specifically regarding the limits of the execution of PI prevention protocol in the COVID-19 group leading to their findings. The significant differences in PI staging are indicative of the acute onset of the wounds in the COVID-19 group.

The mean hospital stay was significantly longer in the COVID-19 patient group. This may be explained by the effort to provide hospital beds for expected COVID-19 patients by discharging non-COVID patients from hospitals and admitting mainly severe cases.

This research points out again the importance of PI prevention and delineates the restrictions and limitations imposed by the pandemic.

Reviewer #2: This is a retrospective observational analysis of pressure injuries during the first wave of COVID-19 in Spain. However, it seems split between trying to describe individual level factors leading to pressure injury (COVID vs non-COVID) and systems level factors (overcrowding and administrative challenges during a COVID wave). For the COVID versus non-COVID analysis it isn't clear to me why patients with pre-existing pressure injuries (the majority of the cases) were included in the analysis. There are many sources of bias for this based on the age structure of COVID versus non-COVID patients and where they were admitted from (home vs subacute care). Also the pre-existing ulcers in COVID patients were likely unrelated to COVID since patients with severe disease usually present to the hospital within a few days to a week of exposure. If the authors want to focus in on the differences in pressure injuries arising from covid versus non-covid, I would only analyze incident cases during the surge, and also exclude non-covid cases that were admitted to the hospital prior to March 1, 2020 because that also adds in a lead-time bias.

However the authors also seem interested in pressure injury incidence as an "indicator of the quality of care in acute settings" during the COVID surge which is also an interesting questions. However for that question, I think an analysis of a March-June 2019 vs March to June 2020 would be a more valid way to assess how systems level challenges affected pressure injuries.

Additional notes:

Introduction

Line 69 - I would also add that isolation precautions have been associated with increased rates of pressure injuries

Methods

Line 108 - I'm still unclear on what an episode of care represents. If patients had multiple episodes were those separate admissions, or had the initial injury resolved but then a second one appeared?

Line 118 - were patients who were first admitted prior to March 1 excluded? There could have been non-COVID patients who had extended inpatient stays prior to March would be a kind of lead-time bias since COVID wasn't present in Spain prior to then.

Results

Line 222 - How was an injury infection defined as opposed to wound colonization?

Line 229 - All open ulcers are colonized with bacteria. Does "microbiological analysis" just mean that some were superficially swabbed for culture (which is irrelevant)? Or were these deep tissue surgical biopsies that are actually clinically relevant?

Line  233 - Again, were these cultures superficial swabs or deep tissue biopsies? Superficial swabs do not often reflect the true etiology of wound infections (https://pubmed.ncbi.nlm.nih.gov/1523451/) and I'd take this portion out of the manuscript, unless they were deep cultures.

Minor

line 79 - It should be "chronic diseases" not "chronical diseases"

6. PLOS authors have the option to publish the peer review history of their article (what does this mean?). If published, this will include your full peer review and any attached files.

Reviewer #1: **Yes: **Moris Topaz, M.D., Ph.D.,

Reviewer #2: No

---

## [Author Response · Author response to Decision Letter 0]

9 Nov 2021

Thank you for submitting your manuscript to PLOS ONE. After careful consideration, we feel that it has merit but does not fully meet PLOS ONE’s publication criteria as it currently stands. Therefore, we invite you to submit a revised version of the manuscript that addresses the points raised during the review process.

RESPONSE: Thank you to the editorial staff and reviewers for their recommendations that will increase the quality and interest of the manuscript; as well as, for the opportunity to send a revised version of the manuscript.

Both reviewers' comments are attached below. Both had differing recommendations whether to accept the paper or not. However, both are in agreement that there is significant bias due to patient selection. And both have suggestions on how to improve this manuscript. Since methodology is the major set back of this submission, it could be that the right choice should be to remove this submission, rewrite the paper focusing on the correct patient population. The reviewers seem to prefer the acute pressure injuries as the group to focus on.

RESPONSE: The authors revised and rewrote the paper following methodological suggestions made by both reviewers (homogenous group).

I still have my bias in also including the chronic wounds in a separate analysis, focusing on issues that were not even mentioned, such as how many of those with deep wounds were offered an operation to alleviate their problem.

RESPONSE: CRD were reviewed with the aim of collecting patients who required intervention by dermatology or plastic surgery. Taking into account the suggestion, the outcomes have been incorporated into the new manuscript. 

I leave it up to the authors to make a decision whether to retreive their paper and submit a revised paper focusing on a more limited homogenous group, or to try and revise their paper taking advantage of the suggestions made by both reviewers.

RESPONSE: We, the authors, have reanalyzed the data following the suggestions of the reviewers and we submit a rewritten paper, including results and discussion sections, and the addition of new tables and figures

Reviewers' comments:

Reviewer #1:

Pressure injuries (PI) are a global health issue. The authors describe and compare epidemiological and clinical features of PIs in COVID-19 patients and patients admitted due to other causes in Internal Medicine Units during the first wave of COVID-19 pandemic.

This study has a substantial inherent bias, performed during the COVID-19 pandemic, that may interfere with the main results. Though it was mentioned in the study limitations, it should be elaborated more specifically regarding the limits of the execution of PI prevention protocol in the COVID-19 group leading to their findings. The significant differences in PI staging are indicative of the acute onset of the wounds in the COVID-19 group.

The mean hospital stay was significantly longer in the COVID-19 patient group. This may be explained by the effort to provide hospital beds for expected COVID-19 patients by discharging non-COVID patients from hospitals and admitting mainly severe cases.

This research points out again the importance of PI prevention and delineates the restrictions and limitations imposed by the pandemic.

RESPONSE: Given your recommendation, a more extended paragraph has been elaborated in discussion related to the limits of the implementation of a PI prevention protocol in the COVID-19 group (Line 306).

Reviewer #2:

This is a retrospective observational analysis of pressure injuries during the first wave of COVID-19 in Spain. However, it seems split between trying to describe individual level factors leading to pressure injury (COVID vs non-COVID) and systems level factors (overcrowding and administrative challenges during a COVID wave). For the COVID versus non-COVID analysis it isn't clear to me why patients with pre-existing pressure injuries (the majority of the cases) were included in the analysis. There are many sources of bias for this based on the age structure of COVID versus non-COVID patients and where they were admitted from (home vs subacute care). Also the pre-existing ulcers in COVID patients were likely unrelated to COVID since patients with severe disease usually present to the hospital within a few days to a week of exposure. If the authors want to focus in on the differences in pressure injuries arising from covid versus non-covid, I would only analyze incident cases during the surge, and also exclude non-covid cases that were admitted to the hospital prior to March 1, 2020 because that also adds in a lead-time bias.

RESPONSE: Done. The inclusion and exclusion criteria have been rewritten to make it clearer that the sample was collected from patients admitted from March 1 to June 1 in the Internal Medicine Service (excluding patients admitted before or after this period).

However the authors also seem interested in pressure injury incidence as an "indicator of the quality of care in acute settings" during the COVID surge which is also an interesting questions. However for that question, I think an analysis of a March-June 2019 vs March to June 2020 would be a more valid way to assess how systems level challenges affected pressure injuries.

RESPONSE: Data have been collected for the same period of the previous year and have been incorporated in the text for discussion (Line 190).

Additional notes:

Introduction

Line 69 - I would also add that isolation precautions have been associated with increased rates of pressure injuries.

RESPONSE: Done. 

Methods

Line 108 - I'm still unclear on what an episode of care represents. If patients had multiple episodes were those separate admissions, or had the initial injury resolved but then a second one appeared?

RESPONSE: Each episode of care represents the assessment, record and treatment of a PI. 

Line 118 - were patients who were first admitted prior to March 1 excluded? There could have been non-COVID patients who had extended inpatient stays prior to March would be a kind of lead-time bias since COVID wasn't present in Spain prior to then.

RESPONSE: The inclusion criteria described in the manuscript have been revised to clarify this point. Only patients admitted from March 1 to June, 1, 2020 were included. 

Results

Line 222 - How was an injury infection defined as opposed to wound colonization?

RESPONSE: According to literature, colonization was defined as the presence of proliferating bacteria without a host response, and therefore does not restrain wound healing. However, infection is related to local and/or systemic host reaction with delayed wound healing. For this reason, the diagnosis of PI infection was combined, although eminently clinical and not only microbiological quantification. In all cases in the sample with infection diagnosis, the nursing staff reported indexes of clinical suspicion of local infection.

Line 229 - All open ulcers are colonized with bacteria. Does "microbiological analysis" just mean that some were superficially swabbed for culture (which is irrelevant)? Or were these deep tissue surgical biopsies that are actually clinically relevant?

Line 233 - Again, were these cultures superficial swabs or deep tissue biopsies? Superficial swabs do not often reflect the true etiology of wound infections (https://pubmed.ncbi.nlm.nih.gov/1523451/) and I'd take this portion out of the manuscript, unless they were deep cultures.

RESPONSE: Although we understand that deep tissue biopsies is the gold standard for microbiological diagnosis of infection in pressure injuries, in clinical practice of our department, cultures are routinely taken by aspiration or superficial swabs. Its bias has been included in the limitations section for the reader´s consideration. 

Minor

line 79 - It should be "chronic diseases" not "chronical diseases".

RESPONSE: Corrected. 

Authors agree with the reviewers’ recommendations. We sent the manuscript as instructed.

---

## [Decision Letter · Decision Letter 1]

10 Dec 2021

PONE-D-21-24374R1Are there differences between COVID-19 and non-COVID-19 inpatient pressure injuries? Experiences in Internal Medicine UnitsPLOS ONE

Dear Dr. Alonso-Sardón,

Thank you for submitting your revised manuscript to PLOS ONE. After careful consideration, we still feel that there is a major deficit to be dealt with before it fully meet PLOS ONE’s publication criteria. Therefore, we invite you to submit a revised version of the manuscript that addresses the points raised during the review process.

We look forward to receiving your revised manuscript.

Kind regards,

Itamar Ashkenazi

Academic Editor

PLOS ONE

Reviewers' comments:

Reviewer's Responses to Questions

**Comments to the Author**

1. If the authors have adequately addressed your comments raised in a previous round of review and you feel that this manuscript is now acceptable for publication, you may indicate that here to bypass the “Comments to the Author” section, enter your conflict of interest statement in the “Confidential to Editor” section, and submit your "Accept" recommendation.

Reviewer #2: (No Response)

2. Is the manuscript technically sound, and do the data support the conclusions?

Reviewer #2: Yes

3. Has the statistical analysis been performed appropriately and rigorously? 

Reviewer #2: No

4. Have the authors made all data underlying the findings in their manuscript fully available?

Reviewer #2: Yes

5. Is the manuscript presented in an intelligible fashion and written in standard English?

Reviewer #2: Yes

6. Review Comments to the Author

Reviewer #2: I appreciate your response to my earlier points and feel that the data presented is now much clearer. However, some of the conclusions don't seem to fit that new data. First, rates of HAPIs were low compared to a year earlier and compared to other literature, which is surprising especially as the age of patients was high with multiple comorbidities. You comment on this is "Although there is a lack of records of specific characteristics of PIs, authors considered that the assessment and management of wounds, as well as decision making of the treatment was probably carried out in situ during the assistance practice and the main reason of registered data loss were care overload and restructuring of nursing staff" which I don't really understand what you are trying to say. Are you saying that numbers of PIs were down because they weren't being accurately recorded? I think this is a major finding and requires more discussion of why you think it happened.

You also place emphasis in the discussion on risk-factors for PI that had a P-value of <0.05 but your analysis has a large amount of comparisons so that value isn't appropriate and needs correction for multiple comparison (eg Bonferroni or Benjamini-Hochberg) or the conclusions should be much more circumspect and this limitation strongly emphasized.

You conclude that "The study shows that HAPIs were more frequent in COVID-19 group during the first wave of COVID-19." but this is only comparing absolute numbers of HAPIs in COVID versus non-COVID. I think you need to compare rates (HAPIs per COVID patient-day versus HAPIs per non-COVID patient-day or HAPIs per COVID admission versus HAPIs versus non-COVID admission) to truly make this conclusion.

7. PLOS authors have the option to publish the peer review history of their article (what does this mean?). If published, this will include your full peer review and any attached files.

Reviewer #2: No

---

## [Author Response · Author response to Decision Letter 1]

22 Jan 2022

Response to Reviewers

Thank you to reviewer #2 for her recommendations that will increase the quality and interest of the manuscript; as well as, thank you to the editorial staff for the opportunity to send a revised version of the manuscript.

Reviewers' comments:

Reviewer #2:

I appreciate your response to my earlier points and feel that the data presented is now much clearer. However, some of the conclusions don't seem to fit that new data. First, rates of HAPIs were low compared to a year earlier and compared to other literature, which is surprising especially as the age of patients was high with multiple comorbidities. You comment on this is "Although there is a lack of records of specific characteristics of PIs, authors considered that the assessment and management of wounds, as well as decision making of the treatment was probably carried out in situ during the assistance practice and the main reason of registered data loss were care overload and restructuring of nursing staff" which I don't really understand what you are trying to say. Are you saying that numbers of PIs were down because they weren't being accurately recorded? I think this is a major finding and requires more discussion of why you think it happened.

RESPONSE: We have clarified this possible confusing point to improve the understanding of the discussion. As we pointed out in the discussion, the prevalence rate is lower than records of the previous year or other national studies, which may sound contradictory after taking into account the serious clinical status of the patients, the high mean age of the sample and pluripathologic condition. Regarding this point, we consider as a likely explanation that overloaded care and overall public health situation may have led to an under-registration of PIs because other registries and/or care were prioritized. Furthermore, in the discussion we tried to highlight that registered LPPs presented an incomplete record of specific characteristics of wounds such as depth, size, wound bed, etc. that were logged in some episodes. Only variables such as staging and location were completed in all episodes.

You also place emphasis in the discussion on risk-factors for PI that had a P-value of <0.05 but your analysis has a large amount of comparisons so that value isn't appropriate and needs correction for multiple comparison (eg Bonferroni or Benjamini-Hochberg) or the conclusions should be much more circumspect and this limitation strongly emphasized.

RESPONSE: Most of the variables analysed are categorical; only some variables, for example age or hospital stay, are continuous variables that allow comparisons by pairs of group means (Bonferroni Test), as we have done. The p-values shown in the tables correspond to the bivariate analysis, as indicated in the methodological section "statistical analysis". Multivariate analysis (logistic regression) was not included in the results because it did not show statistically and clinically significant results.

You conclude that "The study shows that HAPIs were more frequent in COVID-19 group during the first wave of COVID-19." but this is only comparing absolute numbers of HAPIs in COVID versus non-COVID. I think you need to compare rates (HAPIs per COVID patient-day versus HAPIs per non-COVID patient-day or HAPIs per COVID admission versus HAPIs versus non-COVID admission) to truly make this conclusion.

RESPONSE: GACELA-Care® was used for data collection. It uses the "episode of care" as the recording unit. Only total ulcer episode data were available, as figure 1 shows; these data have been exploited in the analysis for the estimation of prevalence and incidence (Results: section 1).

Authors agree with the reviewers’ recommendations. We sent the manuscript as instructed.

---

## [Editor Report · Decision Letter 2]

31 Jan 2022

Are there differences between COVID-19 and non-COVID-19 inpatient pressure injuries? Experiences in Internal Medicine Units

PONE-D-21-24374R2

Dear Dr. Alonso-Sardón,

We’re pleased to inform you that your manuscript has been judged scientifically suitable for publication and will be formally accepted for publication once it meets all outstanding technical requirements.

Kind regards,

Itamar Ashkenazi

Academic Editor

PLOS ONE
---

## [Editor Report · Acceptance letter]

4 Feb 2022

PONE-D-21-24374R2 

Are there differences between COVID-19 and non-COVID-19 inpatient pressure injuries? Experiences in Internal Medicine Units. 

Dear Dr. Alonso-Sardón:

I'm pleased to inform you that your manuscript has been deemed suitable for publication in PLOS ONE. Congratulations! Your manuscript is now with our production department. 

Kind regards, 

on behalf of

Dr. Itamar Ashkenazi 

Academic Editor

PLOS ONE